# Ancestral Origin and Functional Expression of a Hyaluronic Acid Pathway Complement in Mussels

**DOI:** 10.3390/biology14080930

**Published:** 2025-07-24

**Authors:** Umberto Rosani, Nehir Altan, Paola Venier, Enrico Bortoletto, Nicola Volpi, Carrie Bernecky

**Affiliations:** 1Department of Biology, University of Padova, 35121 Padova, Italy; nehir.altan@studenti.unipd.it (N.A.); paola.venier@unipd.it (P.V.); enrico.bortoletto@unipd.it (E.B.); 2Department of Life Sciences, University of Modena and Reggio Emilia, 41125 Modena, Italy; volpi@unimore.it; 3Institute of Science and Technology Austria (ISTA), 3400 Klosterneuburg, Austria; carrie.bernecky@ist.ac.at

**Keywords:** XLINK domain, hyaluronic acid, mussel, *Mytilus galloprovincialis*, HGT

## Abstract

Hyaluronic acid is a key molecule involved in cell adhesion, immune response, and tissue repair, present in vertebrates but rarely detected in invertebrates. In this study, we investigated whether marine mussels could synthesize hyaluronic acid and encode genes enabling its utilization. Through genomic and transcriptomic analyses, we traced two gene loci encoding an extracellular link protein (XLINK), possibly facilitating hyaluronic acid binding. XLINK genes are conserved in the Mytilidae family and actively expressed during mussel development, and they might have been inherited from ancient protostomes or transferred via horizontal gene transfer. Our findings reveal that mussels have the capacity to synthesize hyaluronic acid and a functional pathway to utilize it, representing a rare example of complex trait acquisition. This expands the current understanding of hyaluronic acid biology beyond vertebrates and offers new perspectives on the molecular evolution and functional adaptation of invertebrates.

## 1. Introduction

Hyaluronic acid (HA) is a high-molecular-weight, non-sulfated glycosaminoglycan (GAG) that plays a central role in the biology of vertebrates [1]. Structurally composed of disaccharide units of D-glucuronic acid and N-acetyl-D-glucosamine units, HA is a major constituent of the extracellular matrix (ECM) of vertebrates, where it contributes to tissue hydration, structural integrity, and biomechanical resilience [1]. Beyond its physical properties, HA has been critically involved in different biological processes, including cell proliferation, migration, and differentiation [2]. It facilitates cell–cell and cell–matrix interactions through its binding to specific molecules such as CD44, TSG-6, and RHAMM, among others, thus influencing cellular communication and signaling cascades [3]. Importantly, HA also plays pivotal roles in embryogenesis [4], wound healing, regeneration [5], and immune regulation, where it gained the definition of “stealth molecule”, due to its ability to evade immune recognition [6].

Hypotheses regarding the evolution of HA suggested its late origin during metazoan diversification, likely emerging through the functional diversification of an existing GAG. Interestingly, the advent of HA has been linked to the evolution of separate stem cell niches. In this context, the HA facilitates cell migration by enhancing cellular motility and creating the necessary extracellular space for movement. A notable example is the migration of neural crest cells, where an HA-rich matrix delineates their migratory path [6]. Since HA is known to modulate tumor progression by promoting cell motility, invasion, angiogenesis, and resistance to therapy [7], it has been suggested that its appearance mirrors the evolutionary appearance of malignancies; the metastasizing cancer cell uses an HA-rich pavement for malignant spread [6].

Although the phylogenetic distribution of HA is believed to be chordate- or vertebrate-restricted, a few studies have reported its presence in non-vertebrate species, such as in freshwater mussels [8,9], in the marine mussel *Mytilus galloprovincialis* [10], and in tubeworms [11]. HA synthesis has also been reported to occur in a limited number of bacteria (*Streptococcus pneumoniae*, *Bacillus anthracis*, and *Haemophilus influenzae*, among others) and one yeast species (*Cryptococcus neoformans*), contributing to an increase in pathogenic virulence [12]. Among viruses, only the giant viruses of the *Chlorella* family encode a hyaluronan synthase (HAS) ortholog, which is expressed in early infection when HA is produced and used to establish a productive invasion of host cells [13,14]. Notably, although HAS orthologs have not been traced in non-vertebrate metazoans, the overexpression of HAS2 in *Drosophila melanogaster* was enough to promote the production of HA, suggesting that the presence of HAS is the only requirement for HA production in animals [15].

In addition to HAS, several other proteins interact with HA as receptors, including versican, neurocan, lectican, CD44, RHAMM, and SUSD5, among others, all typically showing a chordate-specific distribution. Likely, they originated from a single ancestor encoding the link module, an extracellular HA binding domain (XLINK) [16]. The *Ciona* XLINK protein does not have the ability to bind HA, but it can bind other GAGs [17]. In contrast, basal chordates such as lancelets (*Branchiostoma* spp.) developed the ability to bind HA and, likely, they also established de novo the biosynthetic pathway to produce HA [17].

Arguably, the absence of HA-associated genes, including HAS and HA receptors, in non-chordate species, poses relevant questions regarding the ability of non-vertebrate species to synthesize and use HA for their own physiology.

To provide a first insight on the possible existence of an HA biosynthetic pathway in invertebrate species as well as the co-option of HA in the biology of these species, we combined genomic, transcriptomic, phylogenetic, and structural approaches to investigate the conservation, phylogenetic distribution, and activation of non-vertebrate XLINK genes. Moreover, we applied a biochemical approach to support the existence and the tissue distribution of HA in tissues of the marine mussel *M. galloprovincialis*.

## 2. Materials and Methods

### 2.1. Data Retrieval and Preliminary Analyses

All the protein sequences available in InterPRO encoding the *extracellular link domain* were downloaded from https://www.ebi.ac.uk/interpro/entry/pfam/PF00193/ (ID: PF00193, accessed on 1 May 2025, Appendix A) and annotated using HMMER v.3.3.2 in combination with Pfam-A v.35.

A total of 18 different genome assemblies belonging to 6 different species of the Mytilinae subfamily were downloaded from NCBI Genomes together with predicted proteins and genome annotations (Appendix A). RNA sequencing datasets of mussel species and *Owenia fusiformis* (annelid) were downloaded from the NCBI SRA database (Appendix A). These datasets referred to mussel developmental stages and tissues (*M. galloprovincialis*, *M. trossulus*, and *M. coruscus*). To download and convert the files into fastq format, we used srahunter v.0.0.7 [18], whereas for read quality trimming, we used fastp v.0.20.1 [19]. All statistical analyses, data manipulation, and visualization steps were performed in R 4.2.3 [20] using the tidyverse [21], ggplot2 [22], ggpubr [23], smplot2 [24], and data.table [25] packages.

### 2.2. Analysis of the XLINK Genomic Loci and Relative Expression Among Mussel Species

Reference mussel genomes were scanned for the presence of the XLINK locus using tblastn searches with the MgXLINK protein sequence (ID: VDH94043.1) as bait. A single *M. galloprovincialis* individual was collected in the southern part of the Lagoon of Venice (Chioggia, VE, Italy, 45°13′34.2″ N 12°16′44.6″ E), immediately transferred to the laboratory, and dissected. The foot tissue was used for high-molecular-weight (HMW) DNA extractions using the Monarch kit (New England Biolabs, Ipswich, MA, USA), resulting in DNA with a size range > 50 kbp, as determined by a Tapestation instrument (Agilent Technologies, Santa Clara, CA, USA). One µg of DNA was used for the preparation of an Oxford Nanopore Technologies (ONT) DNA library, after a step of short fragment removal with the Short Fragment Eliminator Kit (EXP-SFE001, ONT, Oxford, UK). The library was prepared with the Ligation Sequencing Kit V14 (SQK-LSK114, ONT) and sequenced in a Flongle R10.4.1 flow cell, generating a total of 96,970 reads after base-calling and quality trimming. The reads are available in the NCBI SRA database under the accession ID PRJNA1274216.

The RNA-seq reads were used to compute expression levels by mapping them on the corresponding reference genome with the CLC mapper (CLC Bio, Qiagen, Hilden, Germany) with the following parameters, Mismatch cost = 2; Insertion cost = 3; Deletion cost = 3; Length fraction = 0.85; Similarity fraction = 0.85, and expression values were counted as Transcripts Per Million (TPM), to normalize within and between samples.

### 2.3. Sequence Alignment and Phylogenetic Analysis

Full-length mussel XLINK proteins were aligned using MUSCLE [26], and the alignment was further inspected and rendered with CLC Genomics. The protein regions corresponding to the XLINK domains were extracted from the set of XLINK domain-containing proteins obtained from Pfam, and the redundant sequences (>90% identity) were removed using CD-HIT v4.7 [27], resulting in a final set of 3573 sequences. The alignments were performed using the L-INS-i algorithm of MAFFT v7.490 [28], and sites with more than 80% gaps were trimmed using Goalign v0.3.1 [29]. Similarly, sequences with less than 50% of aligned positions were removed. The final alignment is available as Appendix A. Tree reconstructions were done with IQ-TREE v2.2.2.6 [30] with the JTTDCMut + R10 substitution model, which was determined to be most suitable for the analyzed data using ModelFinder [31]. The branch supports were computed using the SH-aLRT test [32] and ultrafast bootstrap estimation [33]. The final consensus tree was uploaded and rendered using the iTOL suite [34].

### 2.4. Structure Prediction of XLINK Proteins

The structures of the full-length MgXLINK proteins (IDs: VDH94043.1 and VDI58096.1) were predicted using the AlphaFold3 server [35]. The structures were aligned, compared, and visualized using UCSF ChimeraX [36]. For comparisons of XLINK domains, the following domain boundaries were used: MgXLINK1 (ID: VDH94043.1), residues 989–1077; MgXLINK2 (ID: VDI58096.1), residues 980–1068; *Acipenser oxyrinchus oxyrinchus* Stabilin-1 (ID: A0AAD8CXX8), residues 2208–2298; murine LYVE-1 (ID: NP_444477.2), residues 29–143; and murine CD44 (ID: NP_001034240.1), residues 24–173. Signal peptides and transmembrane helices were annotated using DeepTMHMM [37].

### 2.5. Quantification of Hyaluronic Acid in Mytilus galloprovincialis Tissues

The different GAGs, hyaluronic acid (HA), chondroitin sulfate (CS), and heparan sulfate (HS), were extracted and purified from the various tissues of adult specimens of *M. galloprovincialis* collected from the Lagoon of Venice (Chioggia, VE, Italy, 45°13′34.2″ N 12°16′44.6″ E). Briefly, after shell removal, the bodies of ten adult mollusks were dissected, and the various collected tissues were pooled and defatted with acetone and treated with papain to digest proteins. After precipitation with ethanol and further centrifugation, the pellet related to the different tissues was dissolved in distilled water, and GAGs were purified by anion-exchange chromatography on a column packed with QAE Sephadex^®^ A-25 anion-exchange resin (Sigma-Aldrich, St. Louis, MI, USA). The collected fractions positive to uronic acid assay were recovered by ethanol precipitation, and the pellet samples were dried, solubilized in distilled water, and further analyzed for single GAG species content.

The HA, CS, and HS content was determined by capillary electrophoresis (CE) equipped with laser-induced fluorescence (LIF) after treatment with specific enzymes able to release the constituent disaccharides, which were further derivatized with a fluorochrome and separated/quantified by CE-LIF [38].

## 3. Results

We inspected the Pfam database searching for proteins encoding the *extracellular link domain* (XLINK, PF00193), and obtained 14,579 hits from 799 taxa (Appendix A). Most taxa were chordates (97.6%); however, 342 hits referred to non-vertebrate species, including 201 hits associated with shotgun metagenomic samples. Excluding chordates, XLINK hits were retrieved from seven species of anthozoans, from bivalves (three species of the family Mytilidae, one Unionidae species, and one Dreissenidae species), as well as from a single tardigrade, an annelid (*O. fusiformis*, 12 hits), and one arthropod species. XLINK hits were also found in bacterial (15), viral (12), and archaea (1) species, with other metagenomic-derived hits not assigned to a given species, thus referring to bona fide viral or bacterial hits.

We also investigated the presence of putative HA synthase enzymes (namely HAS genes) among protostomes, by running iterative blastp searches using the Chlorella virus HAS protein as bait. As a result, only in the *O. fusiformis* genome, we identified four possible orthologs of deuterostome HAS, whereas the confidence levels of the proteins found in other species were below the cut-off (E-values > 10 ×10^−5^) and possibly referred to chitin synthase orthologs (Appendix A).

To further support the existence of an XLINK gene locus in the *M. galloprovincialis* genome, we produced low-coverage sequencing data from an individual mussel collected in the Lagoon of Venice through ONT long reads. We revealed one read covering 18 kb of a first *M. galloprovincialis* XLINK gene reported in the reference genome (MGAL_10B058414, 89.4% of average nucleotide identity, ANI) and a second read covering 11 kb of a second XLINK gene (MGAL_10B015523, 89.2% of ANI).

Irrespective of the partial covering of the reference genes, both deposited and locally produced genomic data supported the presence of two XLINK gene loci in the *M. galloprovincialis* genome.

### 3.1. Two XLINK Domain-Containing Genes Are Conserved in Mussel Genomes

Using blastp searches against the NCBI nr database with the *M. galloprovincialis* XLINK domain-containing protein as bait (NCBI protein ID: VDH94043.1, hereinafter named MgXLINK1), we could retrieve hits exclusively belonging to the genus *Mytilus*.

Only by lowering the identity threshold (>30% of identity over 10% of sequence length) could we retrieve hits of the family Unionidae (freshwater mussels), of the class Gastropoda, and of the phylum Cnidaria. We further evaluated the distribution of XLINK genes in 18 mussel genome assemblies and the related gene predictions, when available. These genomes referred to six species (*M. californianus*, *M. coruscus*, *M. edulis*, *M. galloprovincialis*, *M. trossulus*, and *Perna viridis*, Table 1). In the gene predictions, we identified 14 XLINK hits; in detail, two hits per species were found for *M. galloprovincialis*, *M. trossulus*, and *M. californianus*, three for *M. edulis*, and five for *M. coruscus*. However, the recently published analysis of the *M. coruscus* chromosome-scale genome (with no associated gene annotations) revealed that this species likely possesses only three XLINK genes, with the two additional proteins reported in the previous genome being splicing isoforms. One inconsistent result was found for *M. edulis*, since in the two protein datasets currently available we detected three and two hits, respectively (Table 1). Notably, when haplotype-resolved genomes were available, XLINK gene loci were found in both haplotypes, excluding that these genes are influenced by Presence–Absence Variation (PAV).

Although protein predictions were not available, we could detect two gene loci in *P. viridis*, intriguingly suggesting that the XLINK gene is distributed beyond the genus *Mytilus* (Table 1). To further investigate this aspect, we inspected the transcriptome shotgun assembly (TSA) database of the NCBI, and we could also retrieve transcripts coding for 20 XLINK protein sequences from organisms of the Mytilidae family. The resulting 11 non-redundant hits were obtained from *Septifer virgatus* and *Geukensia demissa*, and from the subfamily Mytilinae (*P. canaliculus*, *P. viridis*, and *Choromytilus chorus*). No additional hits were found when we extended the search to Pteriomorphia.

In sum, extensive database searches showed that the XLINK gene locus is distributed in the Mytilidae family. More divergent proteins are present in a few other mollusk species as well as in anthozoans.

### 3.2. Mussel XLINK1 and XLINK2 Display Distinct Structural Features Compatible with HA Binding

The XLINK proteins of Mytilidae ranged in length from 590 to 1356 residues, after removing splicing isoforms. The XLINK domain is found standalone at the C-terminal protein end in all except one *M. coruscus* protein, in which it is associated with a C1q domain (McXLINK2, Appendix A). All the protein sequences, except the shorter hits, included a signal peptide, suggesting their extracellular localization. Alignment of the full-length proteins of Mytilidae revealed two distinct clusters (Figure 1a) for the XLINK1 and XLINK2 hits (Figure 1b). As the main difference, the XLINK2 hits display a C-terminal transmembrane region and are present exclusively in species from the Mytilinae family. *Mytilus edulis* carries two distinct copies of *XLINK1*, while *Mytilus coruscus* has two different copies of XLINK2. In addition, *Geukensia demissa* possesses four distinct copies of XLINK1 but lacks XLINK2 entirely.

To assess the potential of identified XLINK domains to fold into bona fide HA binding domains, structural prediction of *M. galloprovincialis* MgXLINK1 and MgXLINK2 was performed using AlphaFold3 (Appendix A). The XLINK domains of the two structures were very similar, with an observed root-mean-square deviation of 0.9 Å for confidently predicted regions (pLLDT > 70) (Figure 2a, Appendix A). Consistent with the sequence alignment of the MgXLINK protein clusters, the predicted structures revealed signal peptides located at the N-terminus of both MgXLINK1 and MgXLINK2, as well as a hydrophobic, putative transmembrane helix on the MgXLINK2 C-terminal end (Appendix A).

The predicted structures further suggested that the XLINK domains are topologically similar to already known HA binding domains, including the predicted structure of the *Acipenser oxyrinchus oxyrinchus* Stabilin-1 link domain and the X-ray crystallographic structures of murine LYVE-1 [39] and murine CD44 [40] link domains (Figure 2b–d). Altogether, despite limited sequence similarity, this analysis suggests that the XLINK domains may indeed function as HA receptors.

### 3.3. XLINK Genes Are Transcribed at High Levels During Development in Mytilus Species

We investigated the expression levels of mussel XLINK genes in different developmental stages and different tissues and organs by means of RNA sequencing analysis (Appendix A). Considering three mussel species, in *M. galloprovincialis* the expression levels of MgXLINK1 increased sharply after fertilization, peaked between 8 and 20 h post-fertilization (hpf), and gradually declined at later time points. At 72 hpf, MgXLINK1 was still 2.5 times more expressed than at 4 hpf (Figure 3a). In contrast, MgXLINK2 showed almost no expression at 48 hpf, and it increased to 16 TPM at later time points. The expression profiles of *M. trossulus* MtXLINK1 and MtXLINK2 resembled that of *M. galloprovincialis* MgXLINK1, with the two genes showing very similar temporal expression profiles throughout development (Figure 3b). Both genes increased expression. After fertilization, the expression of both genes increased, peaking between 2 and 17 hpf, and then showing a gradual decline. However, the expression levels of MtXLINK1 were consistently higher compared to MtXLINK2 across all time points, reaching a maximum of 1789 TPM, thus ranking at the 67th position when we ordered the genes by expression level. In *M. coruscus*, only the McXLINK1 gene appeared active: it increased its expression in the first days after fertilization and maintained a stable level from 20 days after fertilization till 60 days, a period roughly covering the pediveliger developmental stage (Figure 3c). For comparison, we considered the expression levels of XLINK and HAS genes in *O. fusiformis*, and we could show that the multiple XLINK genes present in this species were not expressed during development, whereas the expression of a putative HAS ortholog was considerable in the larval stages (Appendix A).

To complement the expression analysis beyond developmental stages, we examined the tissue-specific expression patterns of XLINK genes in adult animals of the same species. For mussels, MgXLINK1 displayed variable expression levels, with hemolymph and gill samples showing the highest levels (Figure 3d). The expression of MgXLINK2 in adult animals was unimportant (<3 TPM). Expression analysis performed in a single replicate per *M. trossulus* tissue indicated that only MtXLINK1 reached detectable expression levels (the expression of MtXLINK2 was irrelevant). MtXLINK1 was expressed in all examined tissues, with hemolymph representing an outlier with 722 TPM (not included in the plot, see Appendix A). Among the plotted tissues, the highest MtXLINK1 expression levels were observed in the gills (around 100 TPM, Figure 3e). *M. coruscus* McXLINK1 gene expression levels were generally similar across most samples, always below 100 TPM (Figure 3f). However, also for this species, two outlier samples were identified, one hemolymph sample (551 TPM) and one mantle sample (430 TPM, both excluded from the plot). The McXLINK2 genes were not expressed in any of the tested samples. Considering *O. fusiformis*, XLINK genes are expressed in the body wall, head, tail, and gut (Appendix A).

### 3.4. The XLINK Domain-Containing Proteins of Protostomes Showed a Patchy Distribution

We extracted all the XLINK domains from the 14,579 proteins retrieved from the Pfam database, adding 11 hits retrieved from the TSA database and referring to Mytilidae species not covered by genomic data. We aligned 3648 non-redundant XLINK domains and purged the alignment from poorly informative sites and sequences. As a result, we obtained 94 informative positions, belonging to 3595 XLINK hits of 505 species (Appendix A). The resulting phylogenetic tree is characterized by multiple clusters, recalling the presence of different protein types that encode the XLINK domain in deuterostomes (Figure 4, https://itol.embl.de/shared/28rDuK6bs4LmS (accessed on 1 July 2025)). We rooted the tree using a cluster of anthozoan hits, considered to be the most basal eukaryotes encoding an XLINK domain-containing gene [17]. Next to this cluster, we found hits of basal chordates, such as lancelets and *Ciona*, together with bivalve species (*Dreissena polymorpha* and *Potamilus streckersoni*, Dreissenidae and Unionidae families, respectively) and 8 out of 12 hits of *O. fusiformis*. Possibly, these sequences are reminiscent of the ancestral XLINK gene found in some anthozoans, which was subjected to duplications and innovations in the lancelets, resulting in up to 32 domains in *Branchiostoma belcheri*. Although most of the *Branchiostoma* spp. hits clustered in this clade, a number of them are spread in the phylogenetic tree, particularly in the hyaluronic acid and proteoglycan link protein (HAPLN) and aggrecan clusters. The most populated cluster (2506 out of 3595 nodes) included aggrecan, neurocan, lectican, HAPLN, stabilin, and TNFP6 proteins, almost exclusive of vertebrates. Lancelets, tunicates, and three *D. polymorpha* hits occupy a single position in this clade (Figure 4, black arrow). The latter hits are close to one hit from hagfish and two hits from fishes of the Gobidae family.

Notably, a group of sequences (all Mytilidae hits and the remaining Unionidae ones indicated by an orange arrow in Figure 4) formed a long-branching cluster close to the CD44 and SUSD5 clusters, suggesting a considerable divergence time. Zooming into this part of the tree, we could show that the nearest sequences are a group of lamprey, salamander, frog, and bony fish (*Acipenser* spp.) hits. Most of the bacterial and viral hits curiously clustered with metagenomic-derived hits, forming a separate clade near the basal XLINK hits and including the single archaeal hit found in the Uniprot database (Figure 4, green arrow). This clade also contains all the hits of the tardigrade *Hypsibius exemplaris* and the sea urchin *Strongylocentrotus purpuratus*, possibly representing an HGT, which may have occurred between bacteria and these species or contaminations. Two Tupanvirus hits clustered in a different clade close to fish hits, leaving open the hypothesis that these viruses acquired the XLINK gene from deuterostomes (Figure 4, gray arrow).

### 3.5. Hyaluronic Acid Is Present in Mytilus galloprovincialis Tissues

To confirm the presence of HA in mussels, which was already reported in a previous study [10], we performed a biochemical quantification of GAGs in six *M. galloprovincialis* tissue or organ pools, obtained from ten adult mussels. HA was detectable in all the analyzed samples, with concentrations ranging from 0.12 ng per mg of tissue to 1.02 ng/mg (Table 2). The foot, gill, gonads, and muscle exhibited relatively lower HA levels compared to the mantle and digestive gland. As expected, the concentrations of CS and HS were substantially higher than HA across every sample, with the mantle showing the highest levels. The relative abundance of HA compared to total GAG content ranged between 0.42% and 2.97%, indicating that HA is a minor but consistently measurable component in these tissues.

## 4. Discussion

Considering a previous study reporting the presence of HA in the Mediterranean mussel [10], we analyzed tissue-specific GAG levels in the same species, *Mytilus galloprovincialis*. The amount of HA originally measured in mussel did not exceed ~10 mg per gram of dry weight [10], i.e., HA levels reasonably sufficient for interactions with biomolecules and for modulation of cell proliferation mollusk flesh having the capacity to aggregate with other biomolecules such as proteins and to modulate cell proliferation as previously demonstrated on in vitro assays [10]. In the present work, the HA amount measured in a single pool of naïve mussels was found to be lower than in the previous quantifications. We demonstrated that mantle and digestive glands are characterized by the highest content of HA compared to the sum of CS and HS, which were observed to be the most abundant GAGs. The relative abundance of HA over total GAGs was higher in the gonad, digestive gland, and adductor muscle, suggesting that in these mussel tissues HA may exert physiological roles in organizing the ECM as well as in the regulation of cell behavior. The possibility that the HA found in the digestive gland of mussels can originate from the diet cannot be excluded, although this is likely not the case for adductor muscle and gonads. The approach we used to quantify HA, based on specific enzymatic treatment, specific derivatization with a fluorochrome, and capillary electrophoresis separation, is able to unequivocally separate HA from other analytes, thus providing a reliable quantification.

The striking presence of HA in mussels implies the existence of an HA biosynthetic pathway in these invertebrates. However, even using sensitive blastp searches among protostome datasets, we could identify putative HAS orthologs only in the annelid *O. fusiformis*. Interestingly, one HAS gene of *O. fusiformis* is considerably expressed in the larvae.

We started with the evidence from public datasets that mussels possess an XLINK domain, and we could support the existence of two XLINK gene loci by producing a low-coverage genome of *M. galloprovincialis*. Furthermore, comparative genomic analyses revealed the conservation of these two loci in all the Mytilinae species analyzed and possibly in other species of the Mytilidae family, but not in other Pteriomorphia. Both mussel genome haplotypes contain the XLINK gene, thus excluding the possibility that XLINK is one of the dispensable genes that greatly contribute to the genetic differentiation and adaptive plasticity of mussel populations [41]. Indeed, extensive genetic introgression has been documented for different Mytilinea species combinations, revealing a complex, yet not fully resolved, evolutionary history [42]. The two XLINK paralogs (XLINK1 and XLINK2) differ in distribution, amino acid sequence, and structure, with all XLINK2 proteins characterized by a C-terminal transmembrane region found restricted to the Mytilinae species. This fact suggests subfunctionalization, with XLINK2 possibly acting as a membrane receptor, and XLINK1 likely secreted in the extracellular space, a situation mirroring the variability of HA binders present in vertebrates [43].

The phylogenetic analysis based on the most informative position of the XLINK domain was effective in dividing the protein types of vertebrates. As previously reported, we could confirm that the origin of deuterostome XLINKs could be linked to an ancestral metazoan gene, still present in anthozoans, subjected to duplication and differentiation starting from lancelets [16]. Accordingly, in addition to a considerable number of gene copies per lancelet species, we also observed the spread of the lancelet hits in the phylogenetic tree, suggesting that lancelet hits predated the extant protein types found in most deuterostomes. Notably, a number of lancelet hits clustered with *Ciona* hits in a basal clade (the *Ciona* hits code proteins unable to bind HA and likely bind a different GAG [17]) and this can mark the de novo evolution of HA.

The phylogenetic tree highlighted the possibility of independent HGTs of XLINK. Possible HGT events are associated with the bivalve *D. polymorpha*, one tick species, and Tupanviruses (Tupanvirus from deep ocean sediment and Tupanvirus from a soda lake). For all these hits, the observed divergence from the nearest vertebrate hits appeared limited, suggesting that these events may have occurred recently. The presence of a tick hit, clustering near the hits of common raccoon dog and desert woodrat, is likely due to the hematophagous behavior of ticks and might underpin functional significance, as shown for similar molecules transferred and integrated into tick genomes [44]. The XLINK hits of the two Tupanviruses, which are protist-infecting giant viruses, clustered near fish hits (*Oryzias melastigma*, *Periophthalmus magnuspinnatus*, and *Iconisemion striatum*), suggesting that the marine environment was the location where the HGT would have occurred. As well, freshwater fishes might represent the source of *D. polymorpha* XLINK. *D. polymorpha* is known to be a freshwater mussel originally distributed in Ukraine and Russia lakes, and this invasive species later spread in different water bodies [45].

Different hypotheses can be envisioned for the origin of Mytilidae, Unionidae, and *O. fusiformis* XLINKs. Both the Unionidae and *O. fusiformis* XLINK hits are clustered in the basal clade near the anthozoan hits and in the cluster with all Mytilidae hits. Notably, this latter cluster also contains two *Nematostella vectensis* hits, whereas the third *N. vectensis* hit is in the anthozoan cluster. Accordingly, one hypothesis would be that the XLINK genes of all protostomes derive from a common ancestor, with extensive gene losses impacting most protostome species. An alternative hypothesis implies a eukaryote-to-eukaryote HGT, which could have introduced a second XLINK type into Mytilidae, *Potamilus streckersoni* (Unionidae), and *O. fusiformis*. This event might have occurred a considerable time ago; this event should be dated back to the early radiation of deuterostomes, in the Cambrian period [46], with the original XLINK protein form possibly equipped with the transmembrane region, which has been subsequently lost in the XLINK2 form, distributed only in the Mytilinae subfamily.

We cannot exclude that the event could have been mediated by an intermediate host, perhaps a virus, having the ability to infect all of the mentioned species.

The presence of XLINK in a single species of each family (except for Mytilidae) might advance the HGT hypothesis compared to the presence of a common ancestor for this XLINK form. However, the presence of a character associated with XLINK that might have contributed to the positive selection of these genes only in the above-described species would support the common ancestor hypothesis. With the available data, we could not validate either hypothesis. Even the possible role of a virus in the transfer of genetic information among ancient bivalves, lancelets, annelids, and fishes could not be identified based on extant viruses, although it might be imagined through paleovirology data showing how relatives of current mollusk herpesviruses were able to infect lancelet and annelid species [47]. It is interesting, anyway, to note how the taxonomic distribution of XLINK genes in non-chordate species mirrored the few reports of the presence of HA, as for *Unionidae* mussels [8], tubeworms [11], marine mussels [10], and Tupanvirus [48]. Structural considerations based on AlphaFold modeling of mussel proteins in comparison with known HA binder structures did not provide definitive evidence of the HA binding ability, nor rejected this possibility. This remains an open question, which may be tackled in future studies through the production of recombinant proteins and proper functional testing.

One last consideration regards the possible roles of HA and of XLINK genes in mussels. Although sample numbers were limited, we reported a strong induction of mussel XLINK genes in early developmental stages of three different mussel species. Interestingly, in *O. fusiformis*, this trend is not present, but one putative HAS gene is expressed in the larvae. Differently, we reported a considerable expression of only a few of the duplicated *O. fusiformis* XLINK genes in tissues like body wall, head, and tail, where, possibly, HA is present. While most of the forms expressed in mussels are XLINK1, *M. trossulus* possesses both genes, expressed with very similar patterns during development, suggesting a common regulatory mechanism. This observation is consistent with what is known regarding XLINK domain-containing proteins in vertebrates, where they play a critical role in mediating HA-dependent cellular processes during tissue formation. In mammals, XLINK domains found in CD44 and TSG-6 proteins guide cell migration and regulate extracellular matrix remodeling through HA binding [49,50]. An intriguing hypothesis is that the strong upregulation we observed for mussel XLINK genes, especially XLINK1, which is a secreted protein like TSG-6, would enable morphogenetic movements, possibly the regulation of ECM. Such similarity suggests that these bivalve XLINK proteins, like their vertebrate counterparts, may facilitate cellular positioning and tissue patterning via HA interaction, although this function remains to be experimentally validated.

## 5. Conclusions

In this work, we provided evidence of the presence of HA in the Mediterranean mussel and of the conservation of the XLINK gene in Mytilinae species as well as in a few other protostome species. Genomic, transcriptomic, and structural results highlighted the conservation of the gene loci and of the expression patterns. This conservation was used to reveal structural similarities with vertebrate counterparts. Despite our analyses, crucial aspects remain to be elucidated. First, the evolutionary paths of XLINK genes remain speculative, with both the HGT and the common ancestor hypotheses being valid alternatives. Perhaps, the identification of the HAS genes in mussel species, as well as other genes involved in the HA pathway, might provide additional arguments. Second, the functional significance of HA as well as the functionality of XLINK proteins in these protostome species requires further investigations, possibly through recombinant protein production and protein immunolocalization. Overall, our analyses revealed an unexpected trajectory with the more conserved evolutionary landscape of metazoan ECM [51].

## Figures and Tables

**Figure 1 biology-14-00930-f001:**
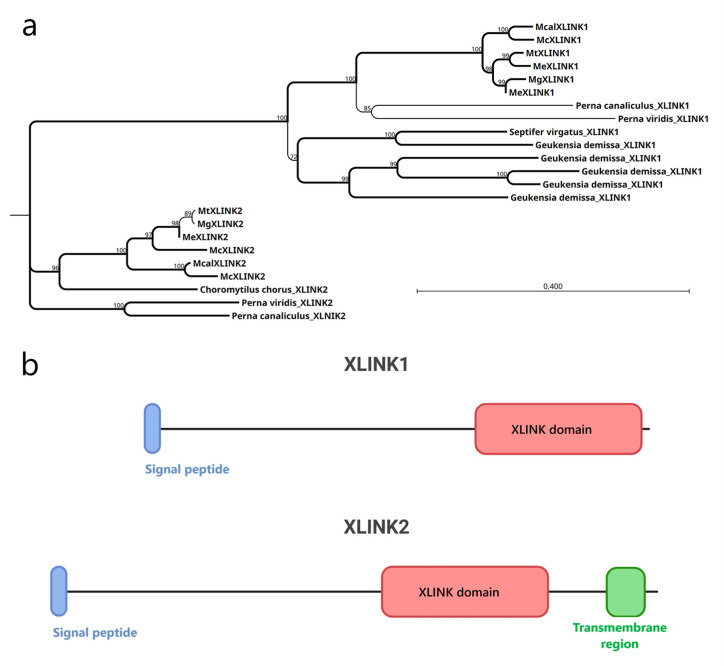
Mytilidae XLINK proteins. (**a**) Phylogenetic tree based on the alignment of 22 XLINK proteins. (**b**) Representation of the typical XLINK1 and XLINK2 protein composition. The signal peptides (blue boxes), the XLINK domains (red boxes), and the transmembrane regions (green boxes) are indicated. *Mt*, *Mytilus trossulus*; *Me*, *Mytilus edulis*; *Mg*, *Mytilus galloprovincialis*; *Mcal*, *Mytilus californianus*; *Mc*, *Mytilus coruscus*. In the phylogenetic tree, percentual bootstrap values are reported for each node; the branches with values higher than 90% are reported in bold.

**Figure 2 biology-14-00930-f002:**
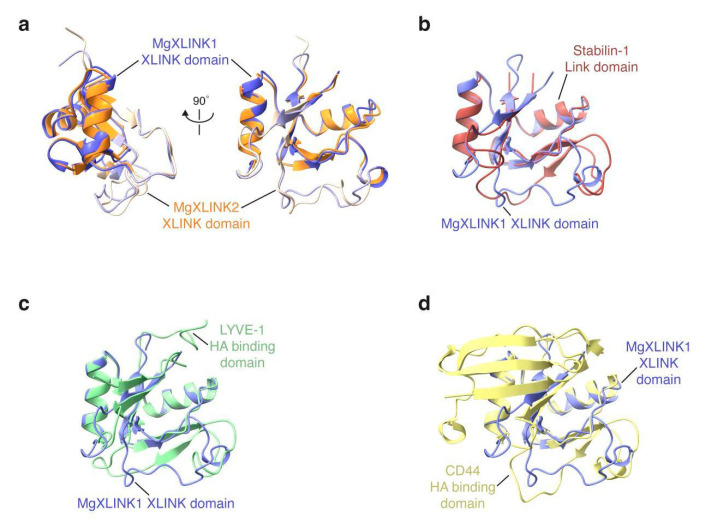
*Mytilus galloprovincialis* XLINK domain structure prediction and comparison to known HA binding domains. (**a**) Overlay of the MgXLINK1 (in purple) and MgXLINK2 XLINK (in orange) domains. Residues that were confidently predicted in both proteins (pLLDT of at least 70) are shown in darker shades. Structural alignment and root-mean-square deviation calculations were carried out using only the confidently predicted residues. (**b**–**d**) Views of the MgXLINK1 XLINK domain (purple) overlaid with several known HA binding domains: (**b**), the *Acipenser oxyrinchus oxyrinchus* Stabilin-1 link domain structural prediction (red); (**c**), the X-ray crystallographic structure of murine LYVE-1 (PDB ID 8ORX) [39] (green); (**d**), the X-ray crystallographic structure of murine CD44 (PDB ID 2JCP) [40] (yellow).

**Figure 3 biology-14-00930-f003:**
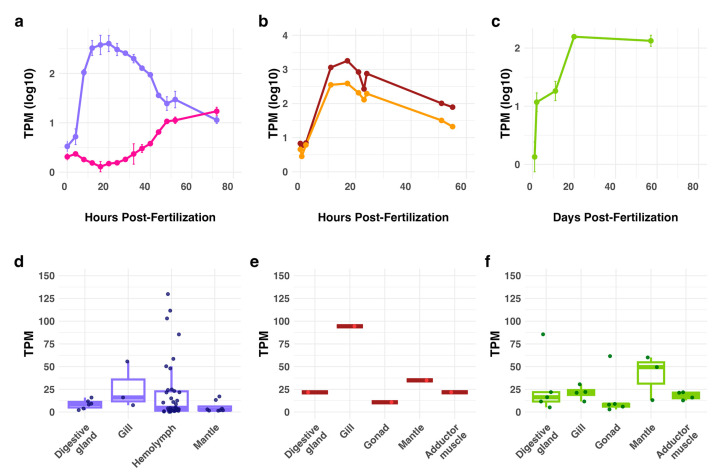
Expression analysis of mussel XLINK genes during developmental stages and across different tissues of three mussel species. Expression levels of XLINK genes during developmental stages were analyzed and reported as TPM in logarithmic scale (log10) in (**a**) *M. galloprovincialis* for genes MgXLINK1 (MGAL10B058414) and MgXLINK2 (MGAL10B015523), (**b**) *M. trossulus* for genes MtXLINK1 (LOC134718524) and MtXLINK2 (LOC134718522), and (**c**) *M. coruscus* for gene McXLINK1 (CAC5355091.1). Expression levels across different individual adult mussel tissues were also assessed for (**d**) MgXLINK1, (**e**) MtXLINK1, and (**f**) McXLINK1. See Appendix A for the related data.

**Figure 4 biology-14-00930-f004:**
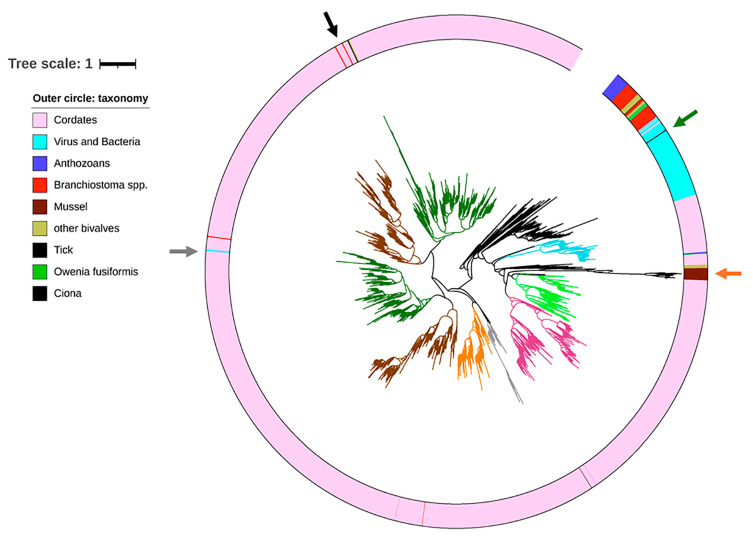
Phylogenetic tree of XLINK domains. The tree is based on 94 informative sites obtained from 3595 aligned XLINK domains retrieved from the Pfam database, plus 11 hits obtained from genomic and transcriptomic data of Mytilidae species. The outer circle encoded the taxonomic classification of these hits according to the color-coded legend. The branch color is informative for the protein types, as retrieved from the available protein annotations (dark green: aggrecan, lectican, neurocan; brown: hyaluronan and proteoglycan link protein; orange: stabilin; gray: tumor necrosis factor-inducible gene 6 protein; pink: LINK domain-containing protein; light green: CD44; light blue: sushi domain-containing protein. 5. The tree is available online in an interactive form at https://itol.embl.de/tree/1471623230252851748985659 (accessed on 1 July 2025).

**Table 1 biology-14-00930-t001:** Summary of mussel XLINK genes. The names of species, the tested genome IDs, the numbers of XLINK loci, the numbers of XLINK predicted proteins retrieved from the genome annotations, and the genome quality level are reported.

Species	Genome ID	No. of XLINK Loci	No. of XLINK Proteins	Genome Quality
*M. galloprovincialis*	GCA_900618805.1 ^1^	2	2	scaffold
	GCA_048414535.1	2	/	primary
	GCA_037788925.1	2	/	primary
	GCA_037788815.1	2	/	alternate
	GCA_025277285.1	2 *	/	primary
*M. edulis*	GCA_905397895.1 ^1^	3	3	scaffold
	GCF_963676685.1 ^1^	2	2	primary
	GCA_963676595.2	3	/	alternate
	GCA_025276775.1	2	/	primary
	GCA_019925275.2	2	/	primary
	GCA_025215535.1	2	/	primary
*M. trossulus*	GCF_036588685.1 ^1^	2	2	primary
*M. coruscus*	GCA_011752425.2 ^1^	6	5	scaffold
	GCA_017311375.1	3	/	primary
*M. californianus*	GCF_021869535.1 ^1^	3 *	2	primary
	GCA_021869935.1	3	/	alternate
*P. viridis*	GCA_037379345.1	2	/	primary
	GCA_018327765.1	2	/	scaffold

^1^ Genomes with predicted proteins available; * one gene was 3′-incomplete. The primary and alternate terms referred to haplotype-resolved genomes for which the two haplotypes were obtained.

**Table 2 biology-14-00930-t002:** Biochemical analysis of glycosaminoglycans (GAGs). Hyaluronic acid (HA), chondroitin sulfate (CS), and heparan sulfate (HS) concentrations were quantified in pooled tissue samples of ten adult *M. galloprovincialis* individuals.

Tissue	ng HA/mg	ng HS/mg	ng CS/mg	Relative Abundance of HA/Total GAGs
Foot	0.12	3.65	24.38	0.42%
Mantle	1.02	27.03	113.63	0.72%
Gonad	0.31	2.88	10.19	2.34%
Gill	0.17	12.98	21.24	0.48%
Digestive Gland	0.89	8.42	20.50	2.97%
Muscle	0.43	3.83	11.94	2.66%

## Data Availability

All the presented data are included as Appendix A or are available in public repositories, as indicated in the text. The sequencing dataset produced for this study is deposited in the NCBI SRA archive under accession ID PRJNA1274216.

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
