# Peer review of "Ancestral Origin and Functional Expression of a Hyaluronic Acid Pathway Complement in Mussels"

_biology, 2025, doi:10.3390/biology14080930_

Round 1
Reviewer 1 Report
Comments and Suggestions for Authors
This study presents compelling evidence for the presence of hyaluronic acid (HA) and its associated XLINK-domain genes in marine mussels (Mytilidae), suggesting acquisition via horizontal gene transfer (HGT) from protochordates.
- The phylogenetic analysis (Fig. 4) places mussel XLINK genes between CD44/SUSD5 clusters and fish/frog sequences, suggesting HGT. However, the long-branch lengths hint at significant divergence, weakening the HGT inference.
- Structural predictions (AlphaFold3) indicate topological similarity of mussel XLINK domains to vertebrate HA binders (Fig. 2), but in vitro or in vivo binding assays are absent. Without experimental confirmation (e.g., HA pulldown assays), claims about HA-binding functionality remain speculative.
- While HA is biochemically detected (Table 2), hyaluronan synthase (HAS) orthologs are notably absent. The manuscript attributes HA production to non-homologous enzymes or bacterial symbionts but provides no data.
- XLINK1 upregulation during early development (Fig. 3) suggests roles in embryogenesis, yet no link is made to specific developmental processes (e.g., cell migration, ECM organization).
- Elaborate on why XLINK2 (transmembrane form) is restricted to Mytilinae. Does this imply subfunctionalization?
Author Response
Reviewer 1
This study presents compelling evidence for the presence of hyaluronic acid (HA) and its associated XLINK-domain genes in marine mussels (Mytilidae), suggesting acquisition via horizontal gene transfer (HGT) from protochordates.
- The phylogenetic analysis (Fig. 4) places mussel XLINK genes between CD44/SUSD5 clusters and fish/frog sequences, suggesting HGT. However, the long-branch lengths hint at significant divergence, weakening the HGT inference.
Thanks for the observation. We are aware of the long-branch feature characterizing this clade of bivalve sequences and we proposed an ancient HGT as the most likely explanation in the original text. However, according also with Reviewer2, we extended this part of Discussion to incorporate an alternative hypothesis, i.e. the possibility of a common ancestor of XLINK in protostomes, followed by extensive gene losses. Both hypotheses cannot be fully supported by data, and remain as such. In the revised version, we tried to improve as much as possible the description of the positions occupied by protostome XLINKs in the tree, to clearly discriminate possible HGTs (likely being involved in the introduction of XLINK in tick, Tupanviruses and one bivalve species) from situations where an HGT is more doubtful. We also included a discussion of O. fusiformis hits (annelid) since they followed a similar distribution. Annelid expression data were also considered in the revised version of the paper, as well as the detection of HAS in this species (see point 3).
- Structural predictions (AlphaFold3) indicate topological similarity of mussel XLINK domains to vertebrate HA binders (Fig. 2), but in vitro or in vivo binding assays are absent. Without experimental confirmation (e.g., HA pulldown assays), claims about HA-binding functionality remain speculative.
We agree on this point, and we clearly stated in the revised version that experimental confirmation experiments will be required to support HA binding (see lines 487-491 and 520-523). Unfortunately, at the present time, we are unable to provide such a validation.
- While HA is biochemically detected (Table 2), hyaluronan synthase (HAS) orthologs are notably absent. The manuscript attributes HA production to non-homologous enzymes or bacterial symbionts but provides no data.
This was just an observation and, certainly, in the absence of detectable HAS genes, the HA production pathway remains an open question. To partially fill this gap, we repeated the search for HAS orthologs with a more sensitive method (psi-blasT) and we could identify putative HAS orthologs in O. fusiformis (as stated in the text, being the only annelid encoding XLINK), but no similar genes were found among bivalves (see lines 179-184). Strikingly, one HAS gene appeared highly expressed in the larvae (see lines 295-298 and the novel Supplementary Figure 3).
- XLINK1 upregulation during early development (Fig. 3) suggests roles in embryogenesis, yet no link is made to specific developmental processes (e.g., cell migration, ECM organization).
- Elaborate on why XLINK2 (transmembrane form) is restricted to Mytilinae. Does this imply subfunctionalization?
Thanks, we have now proposed some explanation for points 4 and 5 in the Discussion.
Reviewer 2 Report
Comments and Suggestions for Authors
The manuscript presents actually interesting, well done and rich in results study. However, the interpretation, especially phylogenetic, should be corrected. At first, I am not convinced that it is really a case of horizontal transfer - the last common abcestor could be invoked. But this possible ancestor lived in the Precambrian (>550 MYA). And molluscs are phylogenetically very far from chordates. Please see my comments in the text. Considering Mytilus as such, perhaps the Authors chould concern a very complicated relationships between the species - for example, M. trossulus has the mtDNA of M. edulis, there are numerous introgressions, etc., etc. Perhaps just a classic character mapping of HA on the molecularly based Tree of Life (or any most recent molecular phylogenetic tree with all the Metazoa) would support interesting results. Any phylogeny with uncomplete set of the taxa may not be reliable. And it has to be understand, that for phylogeny reconstruction we should analyze a sequence which modifications took place at a given time (here so far ago), but these modification are still present: and this is the main problem - consider just saturation...
The circular tree is not easy to interpret, the cladogram should be better described (see the suggestions in your file).

Author Response
Reviewer 2
The manuscript presents actually interesting, well done and rich in results study. However, the interpretation, especially phylogenetic, should be corrected. At first, I am not convinced that it is really a case of horizontal transfer - the last common abcestor could be invoked. But this possible ancestor lived in the Precambrian (>550 MYA). And molluscs are phylogenetically very far from chordates. Please see my comments in the text. Considering Mytilus as such, perhaps the Authors chould concern a very complicated relationships between the species - for example, M. trossulus has the mtDNA of M. edulis, there are numerous introgressions, etc., etc. Perhaps just a classic character mapping of HA on the molecularly based Tree of Life (or any most recent molecular phylogenetic tree with all the Metazoa) would support interesting results. Any phylogeny with uncomplete set of the taxa may not be reliable. And it has to be understand, that for phylogeny reconstruction we should analyze a sequence which modifications took place at a given time (here so far ago), but these modification are still present: and this is the main problem - consider just saturation....
The circular tree is not easy to interpret, the cladogram should be better described (see the suggestions in your file).
Thanks for the positive impression and the suggestions provided. We summarize here the changes we made.
- First, we apologize for the poor readability of the phylogenetic tree. This was because the hyperlink pointed to a figure instead of the interactive tree uploaded in iTol. By using the correct link we hope that this issue could be fully overcome. We also added the required information to Figure 1a (the cladogram), as suggested.
- We corrected the family names, removing unnecessary italics forms. Thanks for highlighting all of them to us.
- The main comment raised to our conclusions pertains to the HGT hypothesis, which we postulated to explain the presence of XLINK in mussels. We have considered your observations and we have now proposed a more balanced discussion. In details, we clearly differentiate between situations where an HGT is the most probable explanation (e.g. Tupanviruses, Dreissena polymorpha and tick XLINKs) from other situations which remain unclear (Mytilidae and Unionidae XLINKs). In this respect, we clarified the number of species having this gene per family, since XLINK appears restricted to individual species in some families (E.g. Owenia fusiformis only, or a single Unionidae species). This supports the HGT, in our opinion. However, in support of your observations, the presence of two Nematostella vectensis hits also in the mussel cluster can support the presence of a common ancestor also for this XLINK form (this element was not highlighted in the original text, thanks). A strong functional constraint may have contributed to the maintenance of this gene only in these few species. We added all these comments to the updated Discussion.
- We also aim to highlight that we built the phylogenetic tree using all the available XLINK hits, not selecting for some taxa. This resulted in a taxonomically-incomplete analysis due to the fact that XLINK is seldom present among protostomes. The presence of XLINK genes in bacteria and viruses is indeed very limited (15 and 12 species, respectively). Therefore, although they represent an ideal vector to move genes across taxa, we couldn't be more specific so far. Of course, a virus able to infect mussels, Owenia fusiformis, unionids, lancelets and gastropods could have easily moved genetic materials between hosts. However, since XLINK is not present in the extant members of the unique viral family that in ancestral time might have had the ability to infect all these hosts (i.e. Malacoherpesviruses), we cannot speculate further.
- We have considered the work done by Prof. Wenne and co-authors regarding mussel speciations to support the presence of extensive genetic connections between these species, resulting in highly plastic genomes (as shown also for the PAV phenomenon we have described starting from 2020 with the sequencing of the Mediterranean mussel genome).
- In the revised version, we have also included some results referring to the search of hyaluronic acid synthase orthologs, since we identified a possible candidate in fusiformis only (among protostomes).
- Finally, according to the presence of two evolutionary hypotheses, we proposed a different title for our manuscript.